# Estimation of calories intake, iron, zinc, and selenium among children of the underprivileged area in Sindh, Pakistan

**Aneel Kapoor**[1]*, **Fizzah Baig**[2], **Naseem Aslam Channa**[3], **Sahar Shafik Othman**[4], **Shahad Abduljalil Abualhamael**[5], **Mukhtiar Baig**[6]

**1** Department of Biochemistry, Peoples University of Medical and Health Sciences for Women, Benazirabad, Sindh, Pakistan, **2** Ziauddin Medical College, Ziauddin University, Karachi, Pakistan, **3** Institute of Biochemistry, University of Sindh, Jamshoro, Pakistan, **4** Department of Family Medicine, Faculty of Medicine, King Abdulaziz University, Jeddah, Saudi Arabia, **5** Department of Internal Medicine, Faculty of Medicine in Rabigh, King Abdulaziz University, Jeddah, Saudi Arabia, **6** Department of Clinical Biochemistry, Faculty of Medicine in Rabigh, King Abdulaziz University, Jeddah, Saudi Arabia

* aneelkapoor@pumhs.edu.pk

## Abstract

### Introduction

Malnutrition is one of the most serious community health issues in developing countries. This study estimated total energy intake, Iron (Fe), Zinc (Zn), Selenium (Se), Calcium (Ca), and Phosphate ($PO_4$) levels among school-going children (aged 13–17 years) of the underprivileged area in Sindh, Pakistan.

### Methods

Children from Mithi City, District Tharparkar, were selected for this cross-sectional investigation. Students from various schools from both genders who fulfilled the selection criteria were selected. A questionnaire was filled, and five ml blood samples were taken to analyze blood parameters. Each participant's estimated nutrient intake (ENI) per day was assessed and matched to the recommended daily allowance (RDA) to determine their micro and macronutrient intake.

### Results

A total of 300 school-going children [150(50%) boys (mean age 15± 0.8 years) and 150 (50%) girls (mean age 14±1.3years)] were included in this study. Total calories (1449±949 Kcal vs. 1245±215 Kcal; p < .001), carbohydrates (138±27 gm vs. 126 ±25 gm; p < .001) protein (47±9.1 gm vs. 44±6 gm; p < .001) was significantly higher among boys compared to girls. In contrast, calcium (1094±105 mg vs. 1144±100; 0.004), phosphate 1050±125 vs. 1148±147; p<0.001), iron (9.2±1.7 mg vs. 10±1.3 mg; p<0.001), and Zinc (7.4±1.8 mg vs. 9.9±1.7 mg; p<0.001) intake was significantly higher among girls than boys. Gender-wise comparison of serum metals in school-going children showed that serum iron was significantly lower among girls than boys (100.86±25.65 µg/dl vs. 78.48±28.66 µg/dl; p<0.001), and no difference was found in serum Zn, Se, and Ca levels. Total proteins were also

**Funding:** The author(s) received no specific funding for this work.

**Competing interests:** The authors have declared that no competing interests exist.

significantly lower among girls than boys (6.48±1.01g/dl vs. 4.87±1.4301g/dl; p<0.001). Serum iron, Ca, and total proteins were significantly lower among girls with normal ranges compared to boys with normal ranges. Total protein was significantly lower among girls below normal ranges than boys with normal ranges (p < .001). The correlation of carbohydrates, protein, and fat with some serum biochemical parameters in school-going children showed that serum Fe was significantly linked with proteins (r = 0.255; p < .0.05).

## Conclusion

Our findings showed a concurrent shortage of macro and micronutrients. The current study also revealed that total energy intake was lower than the RDA and significant Fe, Zn, and Se deficiencies. The findings highlight the importance of measures aimed at improving children's nutritional status.

## Introduction

Pakistan was designated as one of seven countries responsible for about 33% of the world's undernourished population [1]. As a result of its long-term and negative consequences, undernutrition is one of the most important challenges. Undernutrition causes disease and death in children and hinders their physical and cognitive development, scholastic performance, and ability to work in later life [2, 3]. Undernutrition may also cause stunting, wasting, and underweight problems in humans [4]. According to the 2018 National Nutrition Survey in Pakistan, 28.9% of all children are underweight, nearly 40% are stunted, 17.7% are wasting, and 53.5% are anemic [5]. Stunting, wasting, and underweight are more common in rural populations than urban ones, and boys are more stunted, wasted, underweight, and overweight than girls [5, 6].

Malnutrition is one of the leading community health challenges in emerging countries. Usually, it is a silent emergency that produces severe, devastating nutritional consequences in children. Malnutrition affects the majority of children under the age of 15 years. Nearly one in three people worldwide suffer from at least one malnutrition shape, including obesity, underneath-nutrition, and vitamin and mineral deficiencies [7]. The prevalence of nutritional deficiency is at its peak level in South Asia. Malnutrition is among the major health problems, can cause a considerable burden of diseases, and can be the reason for increased mortality in South Asia [8].

Iron is the essential part of hemoglobin (Hb) and works as an oxygen-binding site in Hb. It is also involved in the different oxidative and reductive reactions in cells and works as a cofactor for several enzymes. The myelin sheath synthesis and neurotransmitters also need iron for proper development, release, and generation. Its deficiency may alter the functions of the central nervous system. Iron is the most common micronutrient deficiency worldwide, affecting 1.3 billion children, or 24% of the global population [9, 10].

Zinc is present in plants, animals, and other living forms. It works as a cofactor for the different dehydrogenases. It is essential for the expression of genes, polymerization, and metabolism of DNA and RNA. It influences children, infants, adolescents, and pregnant and lactating women more than adults. It affects the body's growth and development, and the risks of getting infected become higher at deficient levels [9, 11].

Selenium is the integral component of glutathione peroxidase. It is pivotal in the all-inclusive defense mechanism protecting the living organism from free radicals. Its deficiency weakens muscles and causes fatigue, mental fog, a weak immune system, and hair loss [12].

In the South Asian region, the people of Pakistan are predisposed to excessive malnutrition incidence. Malnutrition has been diagnosed in Pakistan for a long time, especially in children and youngsters. It has been reported that nutrient deficiencies are related to poverty and lower literacy rates; almost 1/4 of the population has diminutive access to complete dietary requirements. The prevalence of malnourishment is higher in Sindh province, especially in the remote and neglected areas of Sindh, compared with the other regions of Pakistan [13, 14]. Nutrient intake assessment in growing children is critical component of primary health care. Proper nutrition is fundamental for good health, a healthy immune system, and preventing diseases. Nutrients provide nourishment essential for the growth and maintenance of life [14, 15].

Malnutrition is causing havoc, especially in Pakistan. Therefore, the current study was designed to determine the nutritional status of children and adolescents in rural Sindh. So that appropriate measures to address the issue can be implemented. Tharparker district is a remote, underprivileged area in the province of Sind, Pakistan. The present study was designed to estimate total energy intake, Fe, Zn, and Se levels among school-going children of an underprivileged area in Sindh, Pakistan. The present research work provides awareness to the public, practitioners, parents, and policymakers to design programs to educate and create awareness about the importance of nutrition.

## Methods

### Study design and setting

School-going children aged 13–17 years were selected for this cross-sectional investigation from Mithi City, District Tharparkar, Sindh, Pakistan. Ethical approval was obtained from the Institutional Ethical Committee (Reference No, E1-2016/Ph.D), Institute of the Biochemistry University of Sindh Jamshoro, Pakistan. The conditions laid down in the WMA Declaration of Helsinki 2013 were observed. The participant's health, well-being, and rights were the prior considerations.

The sample collection was started in January 2016 and ended in December 2016. All participants received written information detailing the study's objectives, potential risks, benefits, and the voluntary nature of participation. A written consent form was provided, and participants were required to share this information with their parents/guardians, obtaining their signatures on the consent form. Only children who provided written consent forms signed by their parents/guardians were permitted to participate. Informed written consent forms were given to 918 children, and 300 returned signed ones.

School authorities and the district education officer issued permission to collect data and sampling. Sindh is one of Pakistan's provinces. It is situated in the southeast of the country, and its capital is Karachi. Sindh province has thirty districts. Sindh Province has been divided into many districts over the years due to administrative and demographic shifts. Tharparkar is Sindh's largest district. Geographically, Tharparkar is situated in the southern part of Sindh province in Pakistan and expands about 20,000 sq kilometers. It contains about 200 rural, distant villages burdened by about 90% of the population. Using a 5 percent margin of error, 80 percent power, and mean and standard deviation values from a published Pakistani study [16], the sample size needed to get good statistical power was 36, according to the nQuery online sample size calculator.

## Participants selection

At the time of the current study, Mithi city had two high schools for boys and three for girls. These five high schools were chosen, and study consent forms were disseminated. After getting completed consent forms, study samples were gathered from these high schools. Due to gender-based variations among the different high schools, a stratified sampling technique was employed. The study population was divided into two strata i.e., boys' and girls' schools and then randomly chose samples from each stratum based on their share of the total population in each stratum. This ensured participation from both types of schools and enabled more precise analysis within each grouping.

We chose 300 study subjects from these boys' and girls' schools. There were 150 girls (mean age = 14±1.3years) and 150 boys (mean age = 15± 0.8 years). Our inclusion criteria were 13 to 17-year-olds studying in grades VI to X. According to the exclusion criteria of the present study, children with any infectious disease, autoimmune and chronic diseases (asthma, allergic conditions, tuberculosis, etc.), gastritis, malaria, typhoid fever, and already proven various types of anemia were excluded. The children on any medication were also excluded.

A self-administered questionnaire was designed in the local language to collect data on personal information, general physical examination, and sociodemographic characteristics. The study objectives were explained to all students, and instructions on how to fill out the questionnaire were provided.

## Measurement of estimated nutrient intake (ENI)

Each student's ENI per day was assessed and matched to the RDA to determine their micro and macronutrient intake. A questionnaire was administered to gather data regarding dietary intake through a week's diet history. The food frequency questionnaire was designed with the help of previously published studies [17–19]. The survey was initially formulated in English before undergoing a backtracking process involving two bilingual specialists (English/Sindhi/English). Following that, it was revised by their suggestions. Content validation was limited to the Sindhi translation. A pilot study was conducted by the researchers involving 40 participants in order to validate the survey form's understandability and clarity. The questionnaire language was further modified according to the pilot study responses.

This questionnaire was particularly designed to keep the availability of special types of animal and plant protein foods, fruits, and vegetables of district Tharparkar. The foods taken were estimated by standard pre-measured spoons, plates, and bowls for the amount in grams. All children were informed and practically exercised to complete the data form properly; it included type, weight, time, and amount of meals. A trained team collected the data, which maintained the quality of the data and kept it confidential. The energy contents of the food were analyzed, calculated, and converted for each consumed food/edible to its macro/micronutrients following the national and international databases techniques. One week recall method was used to get the precise nutritional daily intake by dividing the nutrients by seven, and the estimated nutrient intake was equal to the daily intake of each nutrient. Diet records, food frequency questionnaires, and the 24-hour diet recall method are the most commonly used methods in nutrition research. However, the one-week recall method was chosen because many schoolchildren arrive without breakfast due to poverty. All children were physically examined by a qualified physician, and anthropometric measurements were taken. Weight was measured on a bathroom scale calibrated in kilograms (kg) and grams (gm). After removing extra clothes and shoes, the children were weighed with school clothing. Two measurements were taken to the nearest 0.1 kg or 100 gm. The actual weight of the children was taken as the average. Height was measured by applying a measuring tape to the wall and marking

numbers on a blackboard before removing the tape. With barefoot children, measurements were taken from the back of the heels, buttocks, and head touching the wall. The measurements were taken to the nearest 0.5 cm.

## Sample collection

After taking all precautionary aseptic measures, a five ml intravenous blood sample was drawn from all study subjects using a disposable syringe through the vein puncture method in a sitting position. Blood was transferred into plain tubes and kept at room temperature until clotted, and then blood samples were placed in a centrifugation process at 4000 rpm for twenty minutes. The serum was separated and stored at -40 °C in a Biomedical freezer (Sanyo Model: MDF-U5411) for biochemical and metals analysis. The concentration for serum Fe, Zn, and Se was determined by Atomic Absorption Spectrometer Perkin Elmer A800.

## Statistical analysis

SPSS26 was used to analyze the data. The Kolmogorov-Smirnov test was used to check the normal distribution of the data. The frequency and percentages of categorical data were used, while the mean and standard deviation were used for quantitative data. The student t-test was used to examine the statistical difference between the mean values of the two variables. The Pearson correlation test was used to determine the relationship between different variables. The p-value of 0.05 was deemed significant.

## Results

A total of 300 school-going children [150(50%) boys (mean age 15± 0.8 years) and 150(50%) girls (mean age 14±1.3years)] were included in this study. The ENI, like total energy, fats, Fe, and Zn intake in boys and girls, was lower than the RDA. Total calories, carbohydrates, and protein intake were significantly higher among boys, while Ca, $PO_4$, Fe, and Zn were significantly higher among girls. The gender-wise comparison of nutrient intake in study participants is presented in Table 1.

Gender-wise comparison of serum metals in school-going children showed that serum iron was significantly lower among girls than boys. At the same time, no difference was found in serum Zn, Se, and Ca levels (Table 2).

Iron consumption was 87(58%), 60(40%), and 3(2%) normal, low, and high among girls, respectively. Boys had normal, low, and high consumption in 121 (80.7%), 27 (18%), and 2 (1.3%), respectively (Fig 1).

Zinc consumption among girls was 104(69.3%), 38(25.3%), and 8(5.3%) normal, low, and high, respectively. Boys, on the other hand, had 82(54.7%), 63(42%), and 5(3.3%) normal, low, and high consumption, respectively (Fig 2).

Serum iron, Ca, and total proteins were significantly lower among girls with normal ranges compared to boys with normal ranges. Total protein was significantly lower among girls below normal ranges than boys with normal ranges (p < .001) (Table 3).

The correlation of carbohydrates, protein, and fat with some serum biochemical parameters in school-going children showed that serum Fe was significantly linked with proteins (r = 0.255; p < .05). The macronutrients were also significantly correlated with each other. Carbohydrate was significantly correlated with protein (r = 0.471; p<0.05) and fats (r = 0.470; p<0.05), and protein and fates were correlated (r = 0.684; p<0.05) (Table 4).

**Table 1. Gender-wise comparison of nutrient intake among school-going children.**

| Nutrients | *Recommended Daily Allowance RDA | Estimated nutrient intake in boys N = 150 Mean ± SD | Estimated nutrient intake in girls N = 150 Mean ± SD | P-value |
|---|---|---|---|---|
| Energy (Kcal) | 1600–2000 | 1449±949 | 1245±215 | 0.001* |
| Carbohydrates (gm) | 200–300 | 138±27 | 126 ±25 | 0.001* |
| Protein (gm) | 45–55 | 47±9.1 | 44±6 | 0.001* |
| Fat (gm) | 55–70 | 52±9.5 | 50±7.5 | 0.08 |
| Calcium (mg) | 1000–1200 | 1094±105 | 1144±100 | 0.004* |
| Phosphate (mg) | 1000–1200 | 1050±125 | 1148±147 | 0.001* |
| Iron (mg) | 12–15 | 9.2±1.7 | 10±1.3 | 0.001* |
| Zinc (mg) | 10–12 | 7.4±1.8 | 9.9±1.7 | 0.001* |

*RDA was taken from WHO guidelines.

*p<0.05 was taken significant.

## Discussion

The present study showed that both groups had less estimated intake of RDA of energy. Total calories, carbohydrates, and protein intake were significantly higher among boys, while Ca, $PO_4$, Fe, and Zn were significantly higher among girls. Moreover, Fe and Zn intake was lower than the RDA in both groups. It has been reported that girls were more energy deficient among most malnourished children than boys. It usually occurs in younger children because of their increased energy requirements, and it has deleterious effects on their overall health [20, 21].

Like our results, an Indian study also found that girls' calorie, protein, and fat intake was much lower than boys'; however, in contrast to our results, they found less consumption of Ca

**Table 2. Gender-wise comparison of biochemical parameters among school-going children.**

| Serum Metal (Reference Ranges) | Total Children Mean±SD | Boys | Girls | P-Value |
|---|---|---|---|---|
| S. Iron (55–160 µg/dl) N = 121 | 89.27±29.30 | 100.86±25.65 | 78.48±28.66 | <0.001* |
| S. Zinc (Boys 65–118 Girls 59–98) N = 40 | 65.95±16.27 | 67.85±15.32 | 63.95±17.32 | 0.701 |
| ^S. Selenium (70–150 ng/dl) N = 40 | 79±20.73 | 80.5±21.49 | 77.5±20.39 | 0.541 |
| S. Calcium (9-11mg/dl) N = 299 | 9±1.09 | 9.31±1.18 | 8.7±0.93 | 0.91 |
| S. Phosphate (2.7–4.5 mg/dl) | 3.56±1.08 | 3.63±1.20 | 3.50±0.92 | 0.701 |
| Total Proteins (4.5–6.5 g/dl) | 5.81±1.44 | 6.48±1.01 | 4.87±1.43 | <0.001* |
| Total Cholesterol (150–200 mg/dl) | 142.59±29.09 | 142.30±27.10 | 142.87±30.81 | 0.91 |

*p<0.05 was taken significant,

^S = Serum

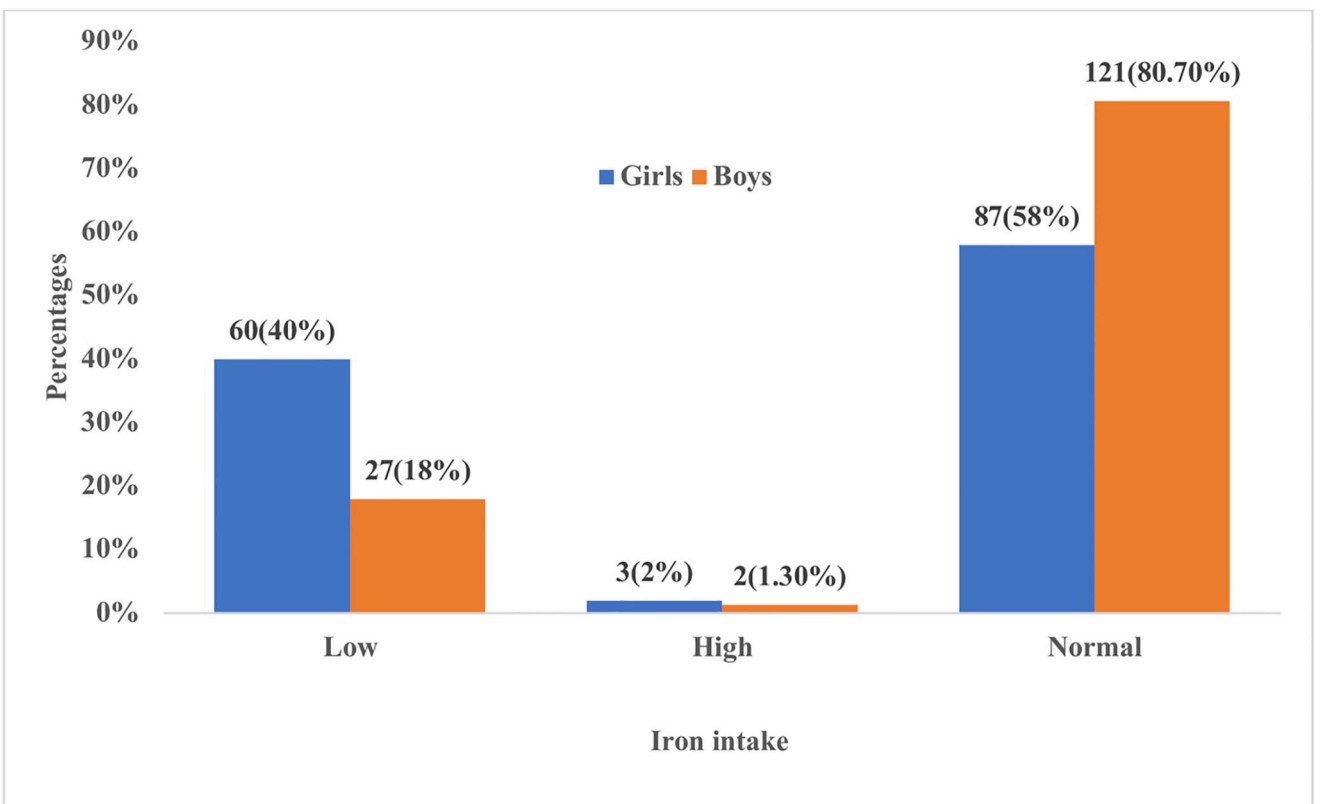

**Fig 1. Iron intake among school-going girls and boys.**

and PO$_4$ among girls [22]. One possible explanation for our findings is that the population in Tharparkar, Sindh, is very poor and cannot afford to buy good sources of nutrients such as meat, fish, eggs, and others, which is why there was a decrease in total calories, Fe and Zn intake in both genders, as well as carbohydrate, proteins, and fats intake in females. However, Fe and Zn deficiencies are higher in boys than girls. These results are comparable to several other studies [21, 23, 24]. However, few studies found no gender-wise difference among children and adolescents in the case of Fe, Zn [25, 26]. According to one report, countries with more than 25% of their population eating a zinc-deficient diet (plant-based diets) are in danger of acquiring zinc deficiency [27]. Because rice and vegetables are the basic diets of our study participants due to their low cost and easy availability, this could be one of the reasons for their low Zn consumption.

There was a 37.5% Se deficiency among our study participants. An Ethiopian study found an even higher incidence of Se deficiency (62%) in school children. The findings contradict Saudi studies that found no significant Se deficiency among normal growing children and adolescents [25, 28]. A New Zealand study found that 22.9 percent of children had low serum Se values [26]. Interestingly, Al-Hussaini [2022] reported that among the low BMI category, 22% of children had Se deficiency [25]. One probable explanation for our research participants' more Se insufficiency is that healthy sources of Se include red meat, seafood, nuts, cereals, poultry, and eggs. Due to their low socioeconomic status, it isn't easy to incorporate these foods into their regular diets. However, in Saudi Arabia, their socioeconomic status is good, and they consume appropriate dietary sources of Se; thus, they do not have a shortfall.

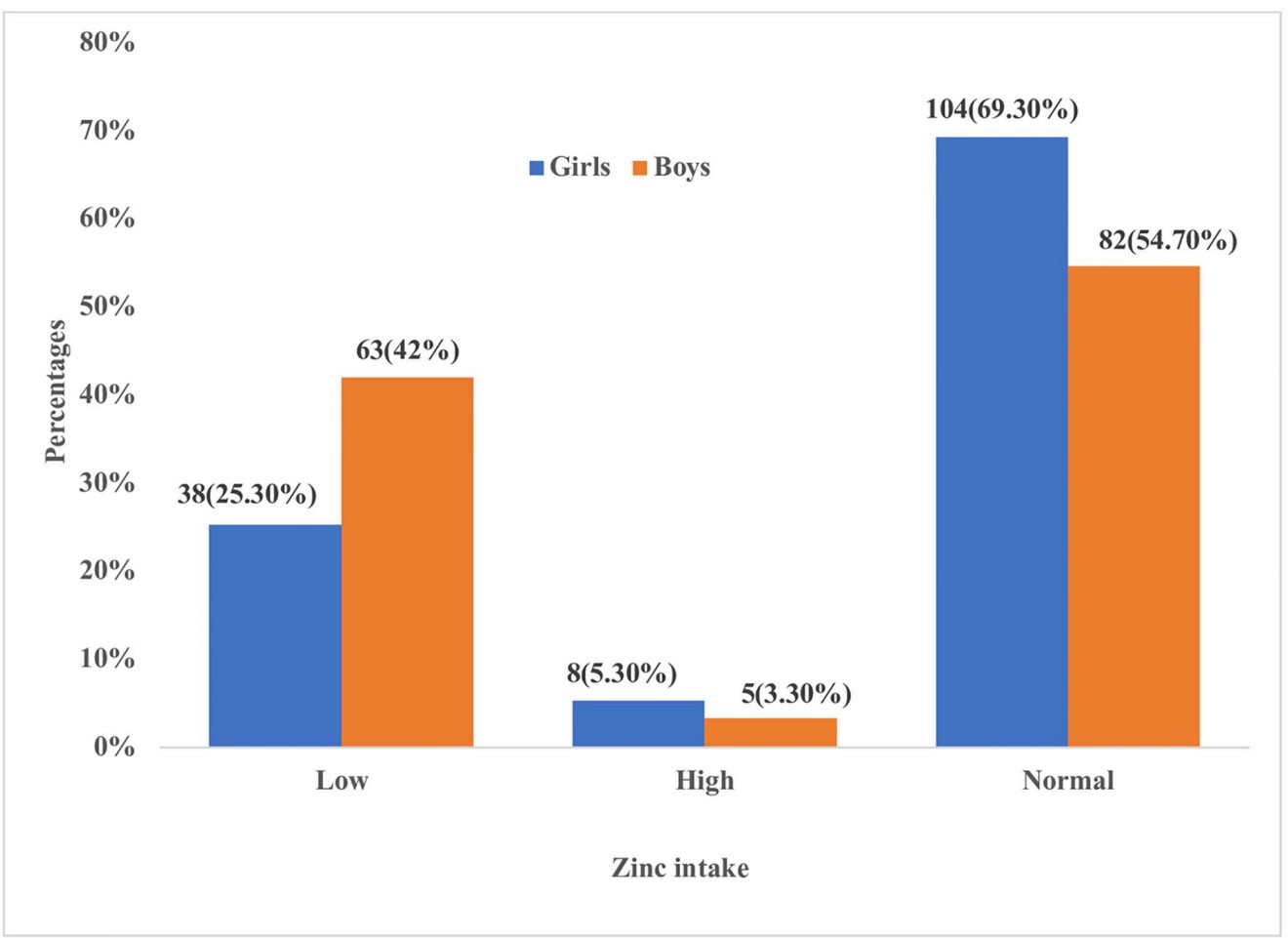

**Fig 2. Zinc intake among school-going girls and boys.**

The current investigation found a deficiency of Zn among 45% of participants. Our findings concur with a Chinese study that found a considerable prevalence of low serum Zn (38.2%) and Fe (24.3%) in children [29]. They also indicated that low Zn was more prevalent than low Fe and that low Zn and Fe levels frequently coexisted. Literate demonstrates that nearly 80% of children were deficient in at least one micronutrient, with 40% deficient in two or more concurrent micronutrients. They further stated a low frequency of Zn insufficiency and no gender differences [30]. New Zealand and Ethiopian studies found that 14.1% and 47% of children had low serum Zn values, respectively [23, 31]. A Saudi investigation found very few cases of zinc deficiency (1%) in children with normal BMIs [25].

The present study found Fe and Zn intake was lower than RDA in both groups, and gender-wise comparison showed Fe and Zn lower intake among boys than girls. Deficiency of Zn is a leading factor for different clinical conditions, including delay in growth, diarrhea, pneumonia, other infections, and disturbance in neuropsychological functions [32].

The estimated nutritional intake (ENI) of boys and girls was considerably lower than the RDA for calories, carbs, proteins, fats, iron, and zinc. The current study showed that girls are more energy deficient than boys. Our findings are comparable to another study [20]. In our study, girls are more energy deficient than boys and have more nutritional deficiencies

**Table 3. Comparison between normal and below normal levels of serum metals among school-going children.**

| Serum Metal (reference ranges) | Boys with normal ranges | | Girls with normal ranges | | P-value | Boys with below normal ranges | | Girls with below normal ranges | | P-value |
|---|---|---|---|---|---|---|---|---|---|---|
| | N | Mean±SD | N | Mean±SD | | N | Mean±SD | N | Mean±SD | |
| Iron (N = 121) (55–160 µg/dl) | 59 | 102.52±24.39 | 46 | 89.32±23.4 | 0.006* | 02 | 52±4.2 | 14 | 42.85±7.03 | 0.099 |
| Zinc (N = 40) (Boys 65–118 Girls 59–98) | 11 | 78.63±10.84 | 11 | 76.27±11.25 | 0.622 | 09 | 54.66±7.5 | 09 | 48.88±9.5 | 0.171 |
| Selenium (N = 40) (70–150 ng/dl) | 12 | 93.66±17.73 | 13 | 89.15±14.74 | 0.494 | 08 | 60.75±4.1 | 07 | 55.85±6.46 | 0.098 |
| Calcium (N = 299) (9-11mg/dl) | 85 | 10.16±0.7 | 66 | 9.6±0.47 | <0.001* | 65 | 8.2±0.4 | 83 | 8.1±0.5 | 0.190 |
| Phosphate (N = 253) (2.7–4.5mg/dl) | 72 | 5.11±1.51 | 24 | 4.89±1.39 | 0.530 | 66 | 3.03±0.28 | 91 | 3.13±0.62 | .223 |
| Total Proteins (N = 255) (4.5–6.5g/dl) | 141 | 6.65±0.73 | 91 | 5.92±0.83 | <0.001* | 08 | 3.52±0.42 | 15 | 2.77±0.33 | <0.001* |
| Total Cholesterol (N = 299) (150–200 mg/dl) | 142 | 145.11±25.1 | 139 | 146.46±28.94 | 0.676 | 08 | 92.50±3.46 | 10 | 93±5.27 | 0.820 |

*p<0.05 was taken significant

because, in our society, parents prefer male children over female children because they consider them as future earning members and caregivers for the family.

Among 13.2% of the study participants, serum iron level was below the normal range. In contrast to our findings, a Nepalian study reported a much higher deficiency of serum Fe (43.4%) among children [33]. Similarly, a recent Saudi study found iron deficiency/depletion in 20% to 25% of Saudi children and adolescents [25]. Ethiopian and Turkish studies reported that female adolescents were times more likely to be anemic than male adolescents [34, 35]. An Indonesian study reported low iron intake compared to the RDA adolescent girls from different socioeconomic [36].

Iron deficiency is the most common nutritional deficiency globally; infants and young children are at high risk. Iron deficiency in a child's body can impair learning ability, cognition, school performance, and conduct, negatively impact the immune system, and be a risk factor

**Table 4. Correlation of carbohydrate, protein, fat intake with serum biochemical parameters among school-going children.**

| Serum Parameters | Carbohydrates ^(r) | Proteins (r) | Fats (r) |
|---|---|---|---|
| Total cholesterol | -0.18 | -0.02 | 0.001 |
| Phosphate | -0.11 | -0.017 | -0.01 |
| Calcium | -0.16 | 0.050 | -0.21 |
| Iron | 0.132 | 0.255* | 0.256 |
| Zinc | 0.228 | 0.111 | 0.146 |
| Selenium | -0.037 | -0.009 | 0.007 |
| Carbohydrates | 1 | 0.471* | 0.470* |
| Proteins | 0.471* | 1 | 0.684* |
| Fats | 0.470* | 0.684* | 1 |

*p<0.05 was taken significant.

^r = Pearson's correlation coefficient

for "attention deficit hyperactivity disorder" and cerebral vein thrombosis [37, 38]. In our study, girls were more deficient in serum iron levels than boys, although their iron intake was higher than boys. The current research was conducted on children between 13 and 17 years old. This is the adolescent age range for girls in which the onset of menstruation occurs. That's why girls have lower levels of iron than boys, and even because of this, they are susceptible to iron deficiency anemia (IDA). Our finding doesn't correspond with an Indonesian study that reported no significant change in Fe deficiency between boys and girls [39].

Malnutrition is a common issue in the rural areas of underdeveloped countries where there are several contributing factors like low literacy rate, high poverty rate, ignorance by their parents, degrading over boys, early marriages, and risk of reproductive morbidity and mortality. Tharparkar district is one of the districts of Sindh with the lowest Human Development Indexing rate. Most of the population lived in villages with low illiteracy and high poverty rates. Among adolescents, girls are more vulnerable, especially in developing countries where they face traditional problems like illiteracy, ignorance by their parents, degrading over boys, early marriages, and risk of reproductive morbidity and mortality.

The concurrent shortage of macro and micronutrients in the present study participants necessitates the establishment of a diversified policy to address the hazard of micro and macronutrient insufficiency among school-aged children; this policy should include 1) medical education campaigns to raise awareness of the benefits of dietary nutrients and a balanced diet among children, adults, and physicians; 2) education about dietary sources rich in micronutrients, and early symptoms of micronutrient deficiency; 3) screening of high-risk groups for malnutrition and supplementation; and 4) fortification of common food staples. As a result, health education is crucial in raising public knowledge of micronutrient-rich food options and healthy dietary practices [25]. The importance of giving relevant health education to mothers should be emphasized because it will enhance their and their children's nutritional status. Additionally, programs to alleviate poverty and improve the nutritional condition of the community's weaker sectors are essential to tackle nutritional deficiencies among mothers and children from low-income homes [40].

## Limitations

The study's sample size is limited to 300 school-age children from Mithi City, District Tharparkar. The results may not be indicative of the larger population, and the conclusions may not be applicable to other locations or age groups.

Nonetheless, this data reflects Pakistan's rural milieu and cannot be compared to prevalence data collected in rural cities in other nations because of lack of facilities, poverty, and very low socioeconomic conditions in Mithi city. Another limitation is that, due to financial constraints, all blood samples from study participants were not tested for all micronutrients. The present study's cross-sectional approach hinders our capacity to demonstrate causal correlations between nutritional intake and malnutrition. Nutrient consumption is estimated using self-reported data obtained through a questionnaire. This method may introduce bias due to mistakes in self-reporting or misinterpretation of portion sizes, which could impair the reliability of the findings. It is admitted that the present study has several types of biases, such as response bias, self-selection bias, communication bias, and sampling bias.

Furthermore, because all the participants live in rural/remote locations, their responses may be prejudiced. This could be due to a lack of time for the interview and the students' shyness. Long-term nutritional status was not seen in the current study, nor was the quality of life assessed; consequently, prospective studies are required to address this issue. Long-term malnutrition status is not observed in the current study, nor is the quality of life assessed;

therefore, prospective studies are required to address this issue. The deficiencies of micro and macronutrients are assessed in the current study using a routine questionnaire and conventional biochemical parameters. Still, many new biochemical and physical markers are emerging that should be analyzed in large-scale study samples.

Despite these limitations, the study sheds light on the dietary issues that disadvantaged children experience in the Tharparkar area, emphasizing the importance of taking steps to enhance their nutritional status.

## Future implications

It is expected that today's children will play major roles in the economy in the future. It is vital to provide them with all the necessary resources for their proper growth and development. Studies should be designed to assess malnutrition in school-aged children, a pressing need in urban and rural Pakistan. Comparable studies should be conducted in various regions of the country to assess the similarities and differences in malnutrition status and biochemical changes associated with food intake. Further long-term follow-up cohort studies are needed to evaluate the effects of food consumed, the development of malnutrition in school-aged children, and the disruption of quality of life. It should also address proper standard international methods of malnutrition diagnosis and management.

## Conclusion

In conclusion, the present investigation showed a concurrent shortage of macro and micronutrients. The current study's findings also revealed significant Fe, Zn, and Se deficiencies. In addition, overall energy intake was below the RDA for both boys and girls in Tharparkar, Sindh province. This altered serum nutrient content in schoolchildren has serious deleterious effects, and policymakers must take prompt steps to improve children's nutritional condition. Further large-scale study findings confirming their assessment are eagerly sought.

## Supporting information

**S1 File. Girls dataset.**
(CSV)

**S2 File. Boys dataset.**
(CSV)

## Acknowledgments

The authors thank all the study participants for their cooperation. It is also acknowledged that this research work is part of the Ph.D. research work of the corresponding author.

## Author Contributions

**Conceptualization:** Aneel Kapoor, Mukhtiar Baig.

**Data curation:** Fizzah Baig, Sahar Shafik Othman, Shahad Abduljalil Abualhamael.

**Formal analysis:** Fizzah Baig, Sahar Shafik Othman, Shahad Abduljalil Abualhamael, Mukhtiar Baig.

**Investigation:** Aneel Kapoor.

**Methodology:** Aneel Kapoor, Naseem Aslam Channa, Mukhtiar Baig.

**Project administration:** Aneel Kapoor.

**Supervision:** Naseem Aslam Channa.

**Writing – original draft:** Aneel Kapoor, Fizzah Baig, Sahar Shafik Othman, Shahad Abduljalil Abualhamael.

**Writing – review & editing:** Naseem Aslam Channa, Mukhtiar Baig.

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
