## [Decision Letter · Decision Letter 0]

19 Jan 2024

PONE-D-23-30249Estimation of calories intake, Iron, Zinc, and Selenium among children of the underprivileged area in Sindh, PakistanPLOS ONE

Dear Dr. Kapoor, 

Thank you for submitting your manuscript to PLOS ONE. After careful consideration, we feel that it has merit but does not fully meet PLOS ONE’s publication criteria as it currently stands. Therefore, we invite you to submit a revised version of the manuscript that addresses the points raised during the review process.

We look forward to receiving your revised manuscript.

Kind regards,

Sajid Bashir Soofi

Academic Editor

PLOS ONE

Journal Requirements:

The name of the colleague or the details of the professional service that edited your manuscript.A copy of your manuscript showing your changes by either highlighting them or using track changes (uploaded as a *supporting information* file).A clean copy of the edited manuscript (uploaded as the new *manuscript* file).

**Additional Editor Comments:**

Thank you for submission of your work. Now the review process is completed, and we have received comments from expert reviewers. Please resubmit your revised manuscript addressing point by point comments of reviewers.

Reviewers' comments:

Reviewer's Responses to Questions

**Comments to the Author**

1. Is the manuscript technically sound, and do the data support the conclusions?

Reviewer #1: Yes

Reviewer #2: Yes

2. Has the statistical analysis been performed appropriately and rigorously? 

Reviewer #1: No

Reviewer #2: Yes

3. Have the authors made all data underlying the findings in their manuscript fully available?

Reviewer #1: Yes

Reviewer #2: Yes

4. Is the manuscript presented in an intelligible fashion and written in standard English?

Reviewer #1: No

Reviewer #2: Yes

5. Review Comments to the Author

Reviewer #1: The ultimate goal of a scientific publication is to expand knowledge, get recognition by citations, recommendations, etc., and to ensure that the study conducted results in improvements in practical activity.

I think the third point received a lot of attention, but the other points received relatively little. As a result, the authors restricted what could have been a highly intriguing study, both in terms of its theoretical potential and its ability to disseminate information.

I regret that the vast bulk of the work's potential has not been realized. Speculating, I would argue that with 10% more effort, you could easily achieve 100% greater results.

I would like to emphasize on the following and authors of this publication should work on those areas.

1. In the abstract, it is imperative that the author provides a report on the significant differences in total calorie intake, carbohydrates, and protein amongst each other, with regards to mean, median, or any other relevant parameter. Additionally, the author should also explicate any significant differences in gender distribution with respect to variables such as Zinc, Calcium, and Iron. It is essential to maintain formality, avoid contractions, and ensure clarity and conciseness while drafting the abstract.

2. The impact of different macronutrients on our health, it's crucial to examine their individual correlation coefficients. Unfortunately, the author overlooked this important detail. By providing the values for carbohydrates, protein, and fat, we could gain a more comprehensive understanding of their respective effects on the dependent variable.

3. The methodology section of the study mentions that it is a community-based study, but it doesn't provide a clear description of the sampling technique used. Ideally, in a study of this nature, a strata sampling or cluster sampling design would be appropriate. It would be helpful if the author could provide more details on how the Sindh Province was divided into different district regions, and the rationale behind the chosen sampling technique. A clear justification of the sampling technique used would enhance the credibility of the study's findings.

4. It seems that the power analysis and sample size used in the study did not adequately explain how the author obtained a 50% response rate from the participants or patients. To address this issue, it may be helpful for the authors to use programs such as PASS or nQuery Statistical software to calculate the necessary sample size and power for the study.

5. There are several chances of bias occurs such response bias, self-selection bias communication bias, drop out bias, etc. how you managed bias?

6. In the methodology section of the paper, the process of validating the questionnaire was not clearly explained. It is crucial to know how the questionnaire was validated in order to establish its reliability and validity. Moreover, the questionnaire was administered in the local language, which raises questions about its accuracy and effectiveness. It would have been helpful if the author had conducted reliability analysis through Cronbach Alpha values to test or re-test the questionnaire's consistency and internal reliability. Therefore, without proper validation and testing, the results based on the questionnaire may not be reliable or generalizable.

7. In the statistical analysis, it is important to ensure that the data follows a normal distribution. Therefore, it is crucial to perform a normality test using the most appropriate method. One of the commonly used methods for normality testing is the Kolmogorov-Smirnov test. Hence, it is essential to check if the author performed the Kolmogorov-Smirnov test or any other appropriate test to verify the normality of the data in the statistical analysis.

8. The author has to rewrite all of the statistical analyses and concentrate on the inappropriate table formatting with respect to their titles as well.

9. it appears that the author conducted their analysis with a focus on gender differences in macro-nutritional deficiencies. However, it's worth noting that the title and objective of the study were coherent with this approach, suggesting that the author intended to investigate gender disparities in nutritional deficiencies. It is important to consider the context of the research when interpreting the analysis, and in this case, the gender-based approach seems to be appropriate. Overall, the coherence between the title, objective, and analysis of the study suggests that the author had a clear research question and methodology in mind, and that their findings are relevant to the topic at hand.

10. Author needs to be improved on the limitation of the study and discussion section.

In conclusion, as I have stated, the authors appear to have worked hard to minimize the work's scope, scientific significance, and practical utility.

If you were under my organizational control, I would advise you to rework some of the arguments, from the theoretical to the data analysis, and rewrite the piece since, if done effectively, it can have an impact in many nations.

But, I merely serve as a reviewer in this capacity, confirming that the piece does, in fact, fit the requirements for publishing.

Reviewer #2: I would like to express my appreciation for the study. The research addresses a significant and timely topic, contributing valuable insights to the field. The authors have done an excellent job in not only conducting a rigorous study but also in presenting their findings with clarity and precision.

One notable aspect of the study is the authors' commendable transparency in addressing the limitations of their research. This acknowledgment reflects their commitment to scholarly integrity and provides readers with a comprehensive understanding of the study's scope and potential constraints.

Furthermore, I would like to highlight the exceptional quality of writing and the overall management of the study. The clarity of language and the logical organization of the content make the study highly accessible to readers.

In conclusion, this study stands out not only for its importance but also for the authors' commendable efforts in communicating their findings effectively and managing the research with a high level of proficiency. I believe this work significantly contributes to the scholarly literature in our field.

6. PLOS authors have the option to publish the peer review history of their article (what does this mean?). If published, this will include your full peer review and any attached files.

Reviewer #1: **Yes: **Muhammad Salman Bashir

Reviewer #2: No

---

## [Author Response · Author response to Decision Letter 0]

15 Feb 2024

Response to reviewers’ comments

Reviewer #1: The ultimate goal of a scientific publication is to expand knowledge, get recognition by citations, recommendations, etc., and to ensure that the study conducted results in improvements in practical activity.

I think the third point received a lot of attention, but the other points received relatively little. As a result, the authors restricted what could have been a highly intriguing study, both in terms of its theoretical potential and its ability to disseminate information.

I regret that the vast bulk of the work's potential has not been realized. Speculating, I would argue that with 10% more effort, you could easily achieve 100% greater results.

I would like to emphasize on the following and authors of this publication should work on those areas.

Response: We appreciate your comments.

1. In the abstract, it is imperative that the author provides a report on the significant differences in total calorie intake, carbohydrates, and protein amongst each other, with regards to mean, median, or any other relevant parameter. Additionally, the author should also explicate any significant differences in gender distribution with respect to variables such as Zinc, Calcium, and Iron. It is essential to maintain formality, avoid contractions, and ensure clarity and conciseness while drafting the abstract.

Response: Thank you for your comments. We have modified the abstract as suggested. 

“Total calories (1449±949 Kcal vs. 1245±215 Kcal; p<.001), carbohydrates (138±27 gm vs. 126 ±25 gm; p<.001) protein (47±9.1 gm vs. 44±6 gm; p<.001) was significantly higher among boys compared to girls. In contrast, calcium (1094±105 mg vs. 1144±100; 0.004), phosphate 1050±125 vs. 1148±147; p<0.001), Iron (9.2±1.7 mg vs. 10±1.3 mg; p<0.001), and Zinc (7.4±1.8 mg vs. 9.9±1.7 mg; p<0.001) intake was significantly higher among girls than boys.

Gender-wise comparison of serum metals in school-going children showed that serum iron was significantly lower among girls than boys (100.86±25.65 µg/dl vs. 78.48±28.66 µg/dl; p<0.001), and no difference was found in serum Zn, Se, and Ca levels. Total proteins were also significantly lower among girls than boys (6.48±1.01g/dl vs. 4.87±1.4301g/dl; p<0.001).”

2. The impact of different macronutrients on our health, it's crucial to examine their individual correlation coefficients. Unfortunately, the author overlooked this important detail. By providing the values for carbohydrates, protein, and fat, we could gain a more comprehensive understanding of their respective effects on the dependent variable.

Response: Thank you for this suggestion. We calculated the macronutrients' individual correlations and found their significant correlation with each other. Carbohydrate was significantly correlated with protein (r=0.471; p<0.05) and fats (r=0.470; p<0.05), and protein and fats were correlated (r=0.684; p<0.05) (Table 4). 

3. The methodology section of the study mentions that it is a community-based study, but it doesn't provide a clear description of the sampling technique used. Ideally, in a study of this nature, a strata sampling or cluster sampling design would be appropriate. It would be helpful if the author could provide more details on how the Sindh Province was divided into different district regions, and the rationale behind the chosen sampling technique. A clear justification of the sampling technique used would enhance the credibility of the study's findings.

Response: Thank you for your comments. We incorporated details of the sample technique in the method section. There are two high schools for boys and three for girls. These five high schools were chosen, and study consent forms were disseminated. After receiving completed consent forms, study samples were gathered from these high schools. 

As there was gender-wise difference between the various high schools or types of schools, a stratified sample technique was adopted. The study population was divided into two strata, i.e., boys' and girls' schools, and then samples were randomly chosen from each stratum based on their share of the total population in each stratum. This ensured participation from both types of schools and enabled more precise analysis within each grouping.

4. It seems that the power analysis and sample size used in the study did not adequately explain how the author obtained a 50% response rate from the participants or patients. To address this issue, it may be helpful for the authors to use programs such as PASS or nQuery Statistical software to calculate the necessary sample size and power for the study.

Response: Thank you for your comments. We have recalculated the sample size value on nQuery statistical software. Using a 5 percent margin of error, 80 percent power, and mean and standard deviation values from a published Pakistani study (Ref 16), the sample size needed to get good statistical power was 36, according to the nQuery online sample size calculator.

5. There are several chances of bias occurs such response bias, self-selection bias communication bias, drop out bias, etc. how you managed bias?

Response: Thank you for your comments. 

Yes, you are right that there are several chances of biases. 

Managing bias in research is crucial to ensure the validity and reliability of research findings. We adopted the following strategies to address common types of biases:

Response Bias:

 - We ensured clear and unbiased wording in questions, and the questionnaire was presented in the local language.

 - We conducted a pilot test to identify potential sources of bias.

 - We trained the data collector to collect the data 

Self-Selection Bias:

 - we ensured selection bias by randomized participant selection from each stratum.

Communication Bias:

 - We trained the data collectors to collect the data, and the data collectors were neutral.

 - We talked with participants in their language to reduce communication bias. 

Drop Out Bias:

 - We monitored dropout rates and only included those participants who gave consent, filled out the questionnaire, and gave blood samples. Moreover, it was not a follow-up study, so dropout was unimportant. 

 Sampling Bias:

 - We used random sampling methods 

 - We analyzed samples gender-wise. 

 - We have clearly defined the target population and sample frame.

6. In the methodology section of the paper, the process of validating the questionnaire was not clearly explained. It is crucial to know how the questionnaire was validated in order to establish its reliability and validity. Moreover, the questionnaire was administered in the local language, which raises questions about its accuracy and effectiveness. It would have been helpful if the author had conducted reliability analysis through Cronbach Alpha values to test or re-test the questionnaire's consistency and internal reliability. Therefore, without proper validation and testing, the results based on the questionnaire may not be reliable or generalizable.

Response: Thank you for your comments. 

A questionnaire was administered to gather data regarding dietary intake through a week's diet history. The present study questionnaire was designed with the help of previously published studies (Ref No. 17-19). The survey was initially formulated in English before undergoing a backtracking process involving two bilingual specialists (English/Sindhi/English). Following that, it was revised by their suggestions. Content validation was limited to the Sindhi translation. A pilot study was conducted by the researchers involving 40 participants in order to validate the survey form's understandability and clarity. The questionnaire language was further modified according to the pilot study. 

As the reliability was not calculated, we have included a sentence about its generalizability in the limitation section.

7. In the statistical analysis, it is important to ensure that the data follows a normal distribution. Therefore, it is crucial to perform a normality test using the most appropriate method. One of the commonly used methods for normality testing is the Kolmogorov-Smirnov test. Hence, it is essential to check if the author performed the Kolmogorov-Smirnov test or any other appropriate test to verify the normality of the data in the statistical analysis.

Response: Thank you for your comments. We used the Kolmogorov-Smirnov test to check the normal distribution of the data. 

8. The author has to rewrite all of the statistical analyses and concentrate on the inappropriate table formatting with respect to their titles as well. 

Response: Thank you for your comments. We have rewritten a few results. We have modified all titles and the formatting of tables.

9. it appears that the author conducted their analysis with a focus on gender differences in macro-nutritional deficiencies. However, it's worth noting that the title and objective of the study were coherent with this approach, suggesting that the author intended to investigate gender disparities in nutritional deficiencies. It is important to consider the context of the research when interpreting the analysis, and in this case, the gender-based approach seems to be appropriate. Overall, the coherence between the title, objective, and analysis of the study suggests that the author had a clear research question and methodology in mind, and that their findings are relevant to the topic at hand.

Response: Thank you for your comments. 

10. Author needs to be improved on the limitation of the study and discussion section.

Response: Response: Thank you for your suggestions. We have modified the limitations and discussion sections as suggested.

The study's sample size is limited to 300 school-age children from Mithi City, District Tharparkar. The results may not be indicative of the larger population, and the conclusions may not be applicable to other locations or age groups.

Nonetheless, this data reflects Pakistan's rural milieu and cannot be compared to prevalence data collected in rural cities in other nations because of lack of facilities, more poverty and very low socioeconomic conditions in Mithi city. Another limitation is that, due to financial constraints, all blood samples from study participants were not tested for all micronutrients. The present study's cross-sectional approach hinders our capacity to demonstrate causal correlations between nutritional intake and malnutrition. Nutrient consumption is estimated using self-reported data obtained through a questionnaire. This method may introduce bias due to mistakes in self-reporting or misinterpretation of portion sizes, which could impair the reliability of the findings. There are several chances of bias occurring, such as response bias, self-selection bias, communication bias, and sampling bias. 

Furthermore, because all the participants live in rural/remote locations, their responses may be prejudiced. This could be due to a lack of time for the interview and the students' shyness. Long-term nutritional status was not seen in the current study, nor was the quality of life assessed; consequently, prospective studies are required to address this issue. Long-term malnutrition status is not observed in the current study, nor is the quality of life assessed; therefore, prospective studies are required to address this issue. The deficiencies of micro and macronutrients are assessed in the current study using a routine questionnaire and conventional biochemical parameters, but many other new biochemical and physical markers are currently emerging that should be analyzed in large-scale study samples. 

Despite these limitations, the study sheds light on the dietary issues that disadvantaged children experience in the Tharparkar area, emphasizing the importance of taking steps to enhance their nutritional status.

Moreover, we have included a few references in the discussion section.

11. In conclusion, as I have stated, the authors appear to have worked hard to minimize the work's scope, scientific significance, and practical utility. If you were under my organizational control, I would advise you to rework some of the arguments, from the theoretical to the data analysis, and rewrite the piece since, if done effectively, it can have an impact in many nations. But, I merely serve as a reviewer in this capacity, confirming that the piece does, in fact, fit the requirements for publishing.

Response: We appreciate your research-oriented approach. We admit that the present study has many limitations. We are thankful for your excellent suggestions and recommendations for the publication of the present manuscript. We appreciate your positive thinking.

Reviewer #2: I would like to express my appreciation for the study. The research addresses a significant and timely topic, contributing valuable insights to the field. The authors have done an excellent job in not only conducting a rigorous study but also in presenting their findings with clarity and precision.

One notable aspect of the study is the authors' commendable transparency in addressing the limitations of their research. This acknowledgment reflects their commitment to scholarly integrity and provides readers with a comprehensive understanding of the study's scope and potential constraints.

Response: Thank you for the positive comments. 

Furthermore, I would like to highlight the exceptional quality of writing and the overall management of the study. The clarity of language and the logical organization of the content make the study highly accessible to readers.

Response: Thank you for your comments. 

In conclusion, this study stands out not only for its importance but also for the authors' commendable efforts in communicating their findings effectively and managing the research with a high level of proficiency. I believe this work significantly contributes to the scholarly literature in our field.

Response: Thank you for your appreciation.

---

## [Decision Letter · Decision Letter 1]

9 May 2024

Estimation of calories intake, Iron, Zinc, and Selenium among children of the underprivileged area in Sindh, Pakistan

PONE-D-23-30249R1

Dear Dr. Kapoor,

We’re pleased to inform you that your manuscript has been judged scientifically suitable for publication and will be formally accepted for publication once it meets all outstanding technical requirements.

Kind regards,

Prof Sajid Soofi

Academic Editor

PLOS ONE

---

## [Editor Report · Acceptance letter]

14 Jun 2024

PONE-D-23-30249R1 

PLOS ONE

Dear Dr. Kapoor, 

I'm pleased to inform you that your manuscript has been deemed suitable for publication in PLOS ONE. Congratulations! Your manuscript is now being handed over to our production team.

Kind regards, 

on behalf of

Professor Sajid Bashir Soofi 

Academic Editor

PLOS ONE